# SemCEB: A Cardinality Estimation Benchmark
# for Semantic Operators

Andreas Zimmerer
University of Technology Nuremberg
Nuremberg, Germany
andreas.zimmerer@utn.de

Claudius Kühn
University of Technology Nuremberg
Nuremberg, Germany
claudius.kuehn@utn.de

Yang Li
The University of Melbourne
Melbourne, Australia
yang.li.11@student.unimelb.edu.au

Mihail Stoian
University of Technology Nuremberg
Nuremberg, Germany
mihail.stoian@utn.de

Renata Borovica-Gajic
The University of Melbourne
Melbourne, Australia
renata.borovica@unimelb.edu.au

Andreas Kipf
University of Technology Nuremberg
Nuremberg, Germany
andreas.kipf@utn.de

## ABSTRACT

Modern data systems increasingly expose multi-modal large language models as *semantic operators*: SQL operators, including filters and joins, whose predicates are defined by a natural-language instruction. Query optimization in these systems still rests on the same foundations as in traditional databases—plan enumeration and cost models—yet faces new challenges, e.g., a larger plan space and the lack of efficient cardinality estimates. The elevated per-tuple costs of semantic operators make bad plan choices worse by orders of magnitude. Therefore, precise—but also fast and cheap—cardinality estimates for semantic filters and joins are of high importance for optimizing query plans that include semantic operators.

In this paper, we introduce SemCEB, the first benchmark for cardinality estimation over semantic operators, based on a real-world dataset of (semi-)structured text and images with 102 hand-curated, diverse queries spanning a wide range of selectivities, assessing cardinality estimation for semantic filters and joins in isolation. We evaluate sampling-based algorithms and *Semantic Histograms*, a state-of-the-art cardinality estimation algorithm for semantic operators, with respect to their accuracy, cost, latency, and memory overhead. We show that, while sampling is robust across different predicate categories, it does not scale and comes with high costs. Our adaptation of *Semantic Histograms*, on the other hand, is limited in its applicability, and its performance appears sensitive to the predicate category.

**VLDB Workshop Reference Format:**
Andreas Zimmerer, Claudius Kühn, Yang Li, Mihail Stoian, Renata Borovica-Gajic, and Andreas Kipf. SemCEB: A Cardinality Estimation Benchmark for Semantic Operators. VLDB 2026 Workshop: Novel Optimization for Visionary AI Systems (NOVAS).

**VLDB Workshop Artifact Availability:**
The source code, data, and/or other artifacts have been made available at https://github.com/utndatasystems/SemCEB.

## 1 INTRODUCTION AND MOTIVATION

The commoditization of multi-modal large language models (MLLMs) has driven a recent surge in semantic data processing systems that treat MLLMs as first-class citizens in query processing. These systems expose MLLM-powered functionality as *semantic operators* [25]: specialized SQL functions that take an instruction (a "prompt") and process input rows using an MLLM according to their defined semantics. A semantic filter, for instance, retains rows based on a natural-language condition. The following query keeps reviews that mention a product failure, referencing the review text in the `review_text` column; syntax may differ between systems:

```sql
SELECT * FROM product_reviews
WHERE AI_FILTER('The review mentions a product failure. '
    || 'The review is: {review_text}');
```

Similarly, *semantic joins* can be expressed as an `AI_FILTER` predicate over the cross-product of multiple tables, but require—as with classical relational joins—dedicated optimizations to achieve acceptable execution times.

Semantic data processing engines remain, at their core, relational systems. Query optimization still relies on plan enumeration and cost models, but now faces much larger plan spaces and the additional objective of maintaining high accuracy under approximate optimizations [21, 27]. Several optimizers for semantic query plans [19, 21] stress that online cardinality estimation remains a key ingredient for plans containing semantic operators, yet current systems largely sidestep this challenge: they rely on fixed selectivity assumptions [21], simple sampling [27], or avoid using cardinality estimates in plan enumeration by treating semantic operators as opaque, overly expensive operators and only re-order filters locally at runtime and otherwise use static rules for operator placement inside the query plan [19].

**Semantic cardinality estimation matters.** One might expect cardinality estimation to matter less here than in classical systems: since execution is dominated by MLLM cost and latency, it almost always pays to evaluate semantic operators late, after relational filtering has reduced the input [21]. In practice, accurate estimates matter more, not less. First, because the cost of a semantic operator scales with the number of rows it processes, a cardinality estimate is effectively a cost and latency prediction—the basis for budgeting, admission control, and surfacing a price to the user before a query

runs. A wrong estimate is no longer merely a slow plan, but a billing surprise. Second, several optimization decisions hinge on selectivity even under late execution: the *relative* order of multiple semantic filters still matters [34], and ordering by selectivity alone is insufficient because filters differ widely in per-tuple cost; semantic filters can sometimes be approximated by online-trained proxy models [4], whose push-down economics depend on selectivity; and physical operator implementations for semantic joins are often tuned for specific selectivity ranges [29]. Production systems such as Snowflake's AISQL engine treat LLM inference cost as a first-class planning objective and report 2–8× end-to-end speed-ups from better plan choices [19].

To further demonstrate the significance of precise cardinality estimates for semantic operators due to their overall increased costs, we showcase an experiment over a three-relation semantic join query with multiple semantic filters over the SemCEB dataset. The query contains a *products* ⋈ *products* join and a *products* ⋈ *reviews* join. All relations have 2-3 semantic filters that are applied before the join, reducing the joins' input sizes. The *reviews* relation has about 10× more rows than the *products* relation before filters are applied. All predicates are conjunctive, so any filter and join order yields the same logical result but a different pattern of intermediate cardinalities. We enumerated all 96 semantically equivalent execution plans (all filter-order permutations crossed with all binary join orders, disregarding join sides) and recorded, for each plan, per-operator input and output cardinalities and LLM cost in tokens and dollars. The cheapest plan costs $0.119 with 524,136 tokens, while the most expensive costs $2.869 with 13,122,385 tokens—a 24.1× cost difference between plans. Interestingly, the optimal join order first performs a *products* ⋈ *reviews* join because of highly selective filters on the *reviews* relation, resulting in a significant drop in end-to-end execution costs. Using fixed heuristics for join-selectivities of, e.g., 20%, the join order would have been ((*products* ⋈ *products*) ⋈ *reviews*), which would have been much more expensive. While the wrong join order dominates the 24.1× cost increase, our experiment showed that a wrong filter-order alone results in a 1.43× cost increase compared to the optimal plan.

The result of this experiment is depicted in Figure 1, and the experiment itself is available in the SemCEB repository. Despite this being a single (reasonable) query instance, it showcases how the impact of wrong join-/filter-order is elevated by increased operator costs, making precise cardinality estimates important.

**Why classical methods and benchmarks fall short.** Cardinality estimation for semantic operators differs fundamentally from its classical counterpart in three ways that together render existing tools inadequate:

- **Estimation is expensive.** Any sampling-based estimator must evaluate the semantic predicate on a subset of the input using the same MLLM, consuming a non-trivial share of the query's budget before execution begins. Estimation thus becomes a three-way trade-off between accuracy, cost, and latency, rather than a pure accuracy problem.
- **Ground truth is model- and prompt-dependent.** Whether a review "mentions a product failure" is a judgment of a specific MLLM under a specific prompt. There is no single objective

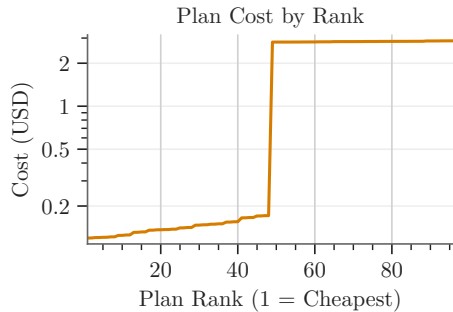

Plan Cost by Rank

**Figure 1: Distribution of virtual LLM inference costs across 96 semantically equivalent execution plans in the SemCEB showcase query. Each plan evaluates the same conjunctive three-relation query but uses a different filter and join order. Costs are computed from virtual-usage cost, excluding cross-plan cache effects. The cheapest plan costs $0.119, while the most expensive costs $2.869, showing a 24.1× cost difference caused solely by execution order.**

selectivity to estimate, only the cardinality that the deployed system would actually observe.
- **Predicates are open-ended and modality-aware.** A semantic predicate is arbitrary natural language, unknown when statistics are collected, so one cannot build histograms over a fixed predicate algebra. The main predicate-independent signal that can be precomputed is an embedding. Embeddings capture equality-like similarity well but encode little about negation, quantitative, temporal, spatial, or lexical conditions; estimator accuracy therefore varies sharply with the *category* of the predicate and the modality of the data. Production experience echoes this: Snowflake reports that the cost and selectivity of semantic operators cannot be inferred from historical statistics or column distributions, leaving them black boxes to the optimizer [19].

While much effort has gone into *executing* semantic operators efficiently—via model cascades [20, 26], batch processing [6], and plan optimization [21, 27]—*estimating* their cardinality has received little attention. Classical benchmarks such as JOB [16], STATS-CEB [9], and Cardbench [3] assume an objective ground truth and a fixed predicate algebra, and account for neither the cost of estimation nor the modality of the data. SemBench [15], the only benchmark dedicated to semantic query processing, targets the cost/quality trade-off of *executing* semantic operators rather than estimating their cardinality. The nascent approaches to semantic cardinality estimation itself—SemStats [34], Semantic Histograms [32], and embedding-aware estimators such as Unify's importance sampling weighted by embedding distance to the query [33]—are each evaluated only within their own system on workloads of their authors' design. This makes them impossible to compare directly and underscores the need for a shared benchmark.

We close this gap with SemCEB, a cardinality estimation benchmark for semantic operators. We make the following contributions:

(1) **A benchmark and workload.** We design a benchmark focused specifically on cardinality estimation for semantic filters and joins in isolation, consisting of 102 hand-written

queries (62 filters and 40 joins) that cover a wide range of predicate properties and modalities, and evaluate estimators along three axes: accuracy, cost, and latency.

(2) **A multi-modal dataset with semantic statistics.** We derive a two-table, multi-modal dataset from Amazon product reviews with 45k products and 936k reviews, including semi-structured text and image columns, precomputed embeddings, and a BERTopic-based measure of semantic skew to characterise estimator difficulty.

(3) **An initial evaluation of semantic estimators.** We evaluate sampling-based estimators and an adaptation of Semantic Histograms [32] on SemCEB, surfacing the fundamental trade-offs between estimation accuracy, cost, and latency, and highlighting how estimator performance varies across predicate categories and modalities.

## 2 RELATED WORK.

There are plenty of recently developed semantic query processing systems, both in industry ([4, 19]) and academia (e.g., [5–7, 12, 14, 20, 25, 28, 31]). While these systems mostly focus on executing semantic operators efficiently and only a few on query plan optimization ([14, 27]), their cardinality estimations rely on naive uniform samples.

Recently, *SemStats* [34] and *Semantic Histograms* [32] both proposed different ideas towards the problem of cardinality estimation for semantic operators and both come with their own limitations, but comparing them is difficult because they use their own workloads. Although *Semantic Histograms* target single-column predicates over image data, the authors argue that its applicability can be extended to other modalities in the future.

Traditional database operators dispose of well-established cardinality benchmarks, e.g., JOB [16], STATS-CEB [9], and Card-bench [3], that aim to stress the query optimizer, particularly when (real-world) correlation is present. Apart from traditional structures such as most common values and histograms, along with sampling-based join size estimation methods [2, 17], there has been a flourishing line of work of both data- and workload-driven learned estimators [10, 13, 22, 35], yet limited to select-project-join queries.

To this end, Lao et al. [15] realized that semantic data processing systems require dedicated benchmarks because neither existing database benchmarks, nor machine learning benchmarks are suitable for the unique challenges, and hence proposed SemBench, a benchmark assessing cost/quality tradeoff of optimizations for semantic operators.

## 3 THE BENCHMARK

The main contribution of this work is an open-source benchmarking framework, written in Python and accompanied by a dataset and a query workload tailored to cardinality estimation for semantic filters and joins. The code is available on GitHub[1] and provides automatic data downloading, packaged ground-truth cardinalities for the queries, baseline implementations, a demo implementation of a cardinality estimation algorithm, the framework runner, and plotting scripts. The framework is transparent about which LLM and which system prompt are used during evaluation, lending it flexibility and keeping the benchmark future-proof. All runtime

properties—such as the choice of LLM and system prompt, the scale-factors, and algorithm-specific configuration—are specified in a single top-level TOML file.

The benchmark exposes a scale-factor, defined as the number of products with associated reviews. A scale-factor of 10 thus yields 10 randomly sampled products in the `products` table and, given an average of 20 reviews per product, roughly 200 associated reviews in the `reviews` table. Sampling is nested, so that larger scale-factors are guaranteed to subsume the items selected at smaller ones. The framework additionally supports a dedicated join-scale-factor: the ground-truth for join queries must currently be obtained with a nested-loop join, since all known optimizations for semantic joins are approximate and may distort cardinalities for complex predicates. The join-scale-factor follows the same logic as the regular scale-factor but applies only to join queries.

Queries in the benchmark deliberately have simple shapes, containing only a single semantic filter or a single semantic join, but might contain multiple column-references in a predicate. In this initial version, we want to avoid complex interactions between operators and focus solely on cardinality estimation for base-table semantic filters and joins; we may extend the benchmark with more complex query shapes in future work.

The ground-truth cardinality of a semantic operator depends on the specific LLM and system prompt in use. We therefore make the deliberate choice to estimate the cardinality that a system would actually observe, rather than an "objective ground-truth" derived from labeled data, and the framework recomputes this model-dependent ground-truth before every run. Because computing ground-truth cardinalities is expensive, the SemCEB ships with cached values for common state-of-the-art models at several reasonable scale-factors.

### 3.1 Dataset

SemCEB's dataset is based on the Amazon Reviews 2023 dataset [11]. To reduce the overall dataset size, we restricted it to the "Arts, Crafts and Sewing" category. To ensure high-quality products and reviews, we applied the 5-core sampling provided by Hou et al. [11], and we additionally required that the following columns be non-`NULL` and contain strings of at least five characters (to discard empty JSON arrays and objects): `product_title`, `features_json`, `description_json`, `details_json`, `images_json`, `review_title`, and `review_text`. For the `products` table, we extracted each product's "main" image and dropped any product for which no such image was available; the downloaded images are distributed with our dataset, and the `main_image_local` column records the local file path of each image. After these preparation steps, the dataset comprised 45k `products` and 936k associated `reviews`.

**Embeddings.** Embeddings are widely used to optimize the execution of semantic operators (e.g., [4, 26]), as they capture semantic meaning in a fixed-size vector. Being independent of any particular semantic predicate, they can be pre-computed by the query engine for columns containing unstructured data. We expect many cardinality estimation algorithms to rely on embeddings to exploit the semantic similarity between tuples, and therefore include pre-computed embeddings in SemCEB. We use state-of-the-art embedding models—`Qwen3-Embedding-0.6B` [36] for textual columns and Google's `siglip2-base-patch16-224` [30] for both text and

[1]https://github.com/utndatasystems/SemCEB

image columns—the latter providing a shared embedding space across modalities. The embeddings are stored in separate columns within the dataset. The dataset additionally includes a boolean column indicating whether the input text was truncated during embedding computation because of the model's context-length limit. Although such truncation occurs only for the `siglip2` model, it is a common way of handling large inputs and is thus a situation that cardinality estimation algorithms should account for. We emphasize that algorithms in SemCEB are not required to use these embeddings and may instead supply their own, specialized embedding computation.

**Assessing Semantic Skew.** Like traditional database operators, semantic operators suffer from data skew, which is particularly problematic for sampling-based cardinality estimation [18]. Skew in semantic data, however, is difficult to assess precisely: although embeddings capture semantic information, inspecting numeric skew along individual dimensions does not yield a meaningful notion of semantic skew, since this information is typically distributed across many dimensions. We therefore propose a new technique for assessing semantic skew based on BERTopic [8]. The BERTopic pipeline first reduces the dimensionality of the embeddings with UMAP [23] and then clusters them into topics using HDBSCAN [1], thereby grouping the embeddings into semantic categories. Assigning the resulting group labels to each tuple recasts the semantic skew problem as a numeric class-imbalance problem, which can then be analyzed with well-established techniques. This recasting aligns naturally with the semantic operator model and offers a useful high-level view of semantic skewness, but reducing skew to a discrete class-imbalance problem precludes a fully detailed analysis; developing richer techniques for characterizing semantic skewness in datasets is, in our view, a promising direction for future work. We note that the resulting grouping depends on both the model used to compute the embeddings and the hyper-parameters of UMAP and HDBSCAN.

Figure 2 shows the semantic class imbalance of the `products` table based on product descriptions; anecdotally, the largest semantic class can broadly be characterized as "everything needed for jewelry-making". The semantic classes are clearly unevenly distributed, yielding a heavily skewed dataset. While such skew is expected of a real-world dataset, it poses interesting challenges for cardinality estimation algorithms.

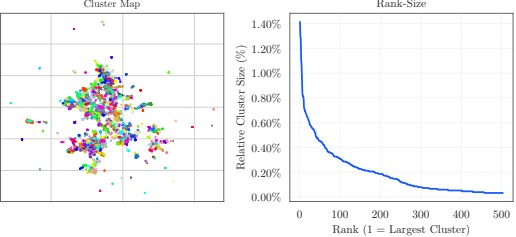

**Figure 2: Semantic skew of the `products` table over textual descriptions of products. For embeddings, the `Qwen3-Embedding-0.6B` was used.**

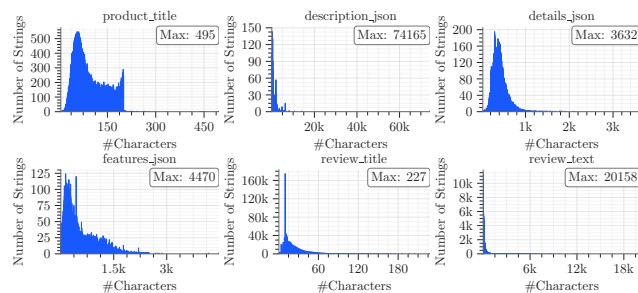

**Figure 3: Distribution of string lengths in (semi-structured) text columns as a proxy for richness of information and to aid choosing an appropriate text-embedding model.**

To assess the informational richness *inside* data, we use the character length in strings as a proxy. The results are depicted in Figure 3.

## 3.2 Queries

Existing benchmarks for semantic operators, such as SemBench [15], target broader aspects of semantic query engines and therefore often employ relatively simple predicates for semantic filters and joins—predicates that lack the complexity one would expect from real-world queries. Prior work has already observed that embedding-based optimizations for semantic operators do not work well for all predicates [26], an experience we share. We argue that, although semantic predicates follow a looser structure than classical relational predicates, certain "categories" of predicates can nonetheless be identified, analogous to the operand types of classical predicates. Our categorization is guided by the intuition of which information is typically encoded in embeddings and which is not, and we conducted experiments to verify these assumptions. Most optimizations for semantic operators perform well on "equality" predicates, where they exploit the high similarity between the predicate and data embeddings, but already struggle with "negation". Quantitative attributes, temporal and spatial information, and lexical information are likewise rarely captured by embeddings. All benchmark queries fall into one or more of these categories.

Predicates may further reference one or several columns. This contrasts with classical predicates, since referencing multiple columns within a single predicate enables rather unusual comparisons, such as *"is the object depicted in {image1} larger than the one in {image2}?"*. The following table reports how many filter and join queries in SemCEB reference a single column versus multiple columns; we classify a join that references exactly one column from each table as a single-column join.

|  | Filters | Joins |
| --- | --- | --- |
| Single-Column Predicates | 37 | 23 |
| Multi-Column Predicates | 25 | 17 |
| Total | 62 | 40 |

During query curation, we ensured coverage of a diverse range of predicate selectivities, shown in Figure 4; the exact selectivities depend on the model used for evaluation.

**Table 1: Categorization of semantic predicates into classes together with examples and how often each category appears in SemCEB. Note that a single predicate might be categorized into multiple classes, for instance** *"The product title asserts a specific quantity [...], but the product details do not confirm or are inconsistent with that quantity"* **is categorized as "Equality", "Negation", and "Ordinal".**

| Category | Sub-Form | Meaning | Filter Example | Join Example | #Preds in SemCEB |
|---|---|---|---|---|---|
| **Equality** | exact-equality
category-membership | "is/is-a" identity
belongs to a class | review is positive
in "knitting" category | products are the same item
same craft category | 52 |
| **Negation** | negation
contradiction | negates an attribute
semantic opposition | no price mentioned
contradicts description | describe different products
same item, opposite verdicts | 23 |
| **Ordinal** | quantitative
degree/intensity | numeric/amount order
subjective strength | rated above 4 stars
strongly positive | A has more items than B
A more detailed than B | 17 |
| **Temporal** | absolute-time
relative-time | fixed point/interval
relative ordering | posted after 2020
a more recent trend | reviewed in same year
A applied before B | 11 |
| **Spatial** | spatial-relation
spatial-extent | position/containment
size/coverage | logo in the corner
fills most of frame | same camera angle
A shown larger than B | 4 |
| **Lexical** | contains/ILIKE
style/topic | keyword present
style/topic fit | mentions durability
matches "rustic" style | keywords appear in B
topic matches category | 10 |

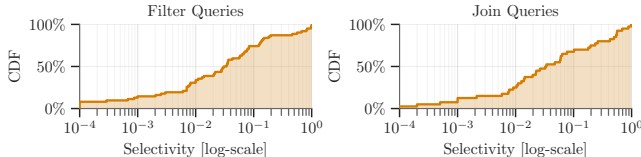

**Figure 4: Selectivity distribution of semantic filters and semantic joins in SemCEB when evaluated with `gpt-5.4-nano`. Filters are executed with scale-factor 1000, joins with scale-factor 100.**

## 4 EVALUATION

We evaluated several approaches within our framework. Owing to the high cost of obtaining ground-truth cardinalities for semantic operators, we evaluated semantic filters at scale-factor 1000 and semantic joins at scale-factor 100.

**Metrics.** We quantify estimation quality using the q-error [24], the symmetric factor by which an estimate deviates from the true cardinality. For example, if the true cardinality is 130, both an estimate of 13 and an estimate of 1300 yield a q-error of 10, since each is off by a factor of 10. Because our workload includes highly selective predicates (Figure 4), the true cardinality $c$ or the estimate $\hat{c}$ can be zero or near-zero, in which case the raw ratio is undefined or explodes. We therefore adopt the standard convention of clamping both quantities at one before taking the ratio:

$$q\text{-error} = \max\left(\frac{\max(\hat{c}, 1)}{\max(c, 1)}, \frac{\max(c, 1)}{\max(\hat{c}, 1)}\right)$$

This assigns a perfect q-error of one to an empty true result that is correctly estimated as empty, and it bounds the penalty for the degenerate cases that would otherwise dominate the metric. The same convention is used in prior cardinality-estimation benchmarks [9], making our filter results directly comparable. Beyond accuracy,

estimation latency and cost are of major importance: we expect many semantic cardinality estimation algorithms to invoke an LLM during estimation, typically incurring substantial cost and latency. Finally, we measure the storage footprint of the additional metadata maintained for cardinality estimation.

**Approaches and Baselines.** We evaluate simple sampling-based estimators with varying sample sizes. Samples are drawn uniformly from the base tables, and the cardinality is estimated by evaluating the filter or join on the sample and extrapolating to the full table size. In addition, we adapted an implementation of *Semantic Histograms* [32] from the original source code. While the original work targets image data only, we extended the code to support textual data; our adaptation nonetheless remains limited to a single referenced column per predicate. We implemented only their KV-cache scanning approach and evaluated neither the Specificity Model nor the ensemble approach, in both cases because of difficulties in transferring the implementation to generic predicates over textual data, and we did not implement KV-cache compression. We did not include SemStats [34] because its implementation was not available to us at the time of writing.

**Results.** Figure 5 depicts the main comparison of the implemented semantic cardinality estimation algorithms across different metrics for filter and join queries, respectively.

Our adaptation of *Semantic Histograms*—like the original—supports only single-column filters, and hence no joins, so that only 31 of the 102 queries in SemCEB are covered by this technique. To ensure a fair comparison, Figure 5 reports two workload executions: one over all 102 queries, and one over the 31 queries supported by *Semantic Histograms*, which allows a direct comparison between the sampling-based approaches and this technique. While the q-error of *Semantic Histograms* is generally substantially higher than even that of a 1% sample—roughly consistent with the results reported by its authors [32]—a single cardinality estimation takes

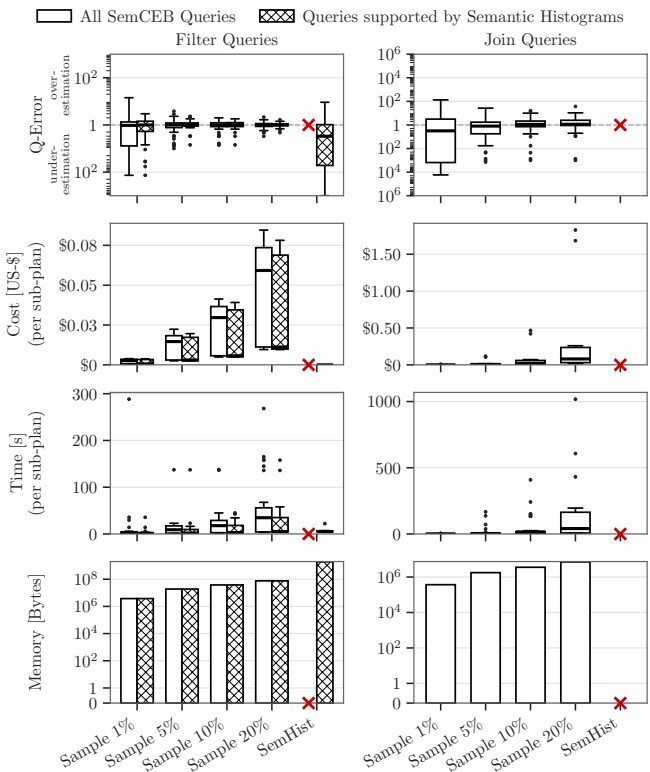

Figure 5: Comparison of evaluated semantic cardinality estimation algorithms—uniform sampling with different sample sizes (`Sample`) and Semantic Histograms (`SemHist`)—across different metrics. Experiments conducted with scale-factor 1000 for filters and 100 for joins using `gpt-5.4-nano` as the LLM.

only around 6 s on average, making it an attractive candidate for queries containing many semantic operators. This is undercut only by the sampling-based algorithm at a 1% sample, which averages 5 s, at the cost of a substantial memory overhead due to its use of KV-caches. We note that we did not implement KV-cache compression for this approach, which might reduce that overhead considerably. Sampling-based approaches produce relatively precise estimates on this dataset already at a 5% sample size; however, the small scale-factor of 1000—chosen to keep costs reasonable—makes it difficult to draw general conclusions about a sufficient minimum sample size. Sampling inherently incurs a significant cost and latency penalty, even more so for join queries, making it a poor candidate for large query plans with many semantic operators.

Figure 6 breaks down the q-error by query category for the sampling-based approach with a 5% sample and for *Semantic Histograms*. On the handful of queries available per category, the q-error for negation and temporal predicates was notably higher for our adaptation of *Semantic Histograms* than for the robust sampling-based approach, suggesting that *Semantic Histograms* struggles with certain predicate types—possibly an artifact of the smaller model used for the KV-cache evaluation.

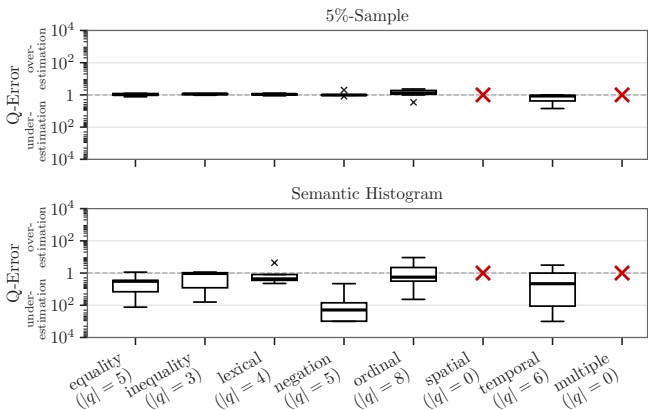

Figure 6: Q-error breakdown for extrapolation from a 5% sample and *Semantic Histograms* across different predicate categories for semantic filters with scale-factor 1000 executed with `gpt-5.4-nano`. Only queries that are supported by both methods are shown. The number of queries that fall within a category is depicted below the category name.

## 5 LIMITATIONS AND FUTURE WORK

While this work focuses on cardinality estimation for semantic filters and joins in isolation, other semantic operators may likewise benefit from precise cardinality estimates. The *semantic group-by* operator, for instance, which assigns input tuples to one of an *a priori* unknown number of groups, could—much like its relational counterpart—exploit an accurate upfront estimate of the total number of groups for resource allocation. We nevertheless excluded semantic group-by: current implementations of this operator remain immature, and the community has yet to converge on a precise definition of its semantics, which makes computing reliable ground-truth cardinalities difficult. Once such a consensus emerges, we expect that group-by queries can be incorporated into SemCEB.

A further deliberate restriction of SemCEB is that it benchmarks cardinality estimation for individual operators in isolation, setting aside interactions with both classical relational operators and other semantic operators. This scoping allows us to isolate the accuracy/cost/latency trade-off that is central to semantic cardinality estimation. We leave the assessment of these estimators' end-to-end impact on hybrid query plans to future work.

We hope SemCEB fosters further research on the problem of semantic cardinality estimation.

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
