# OpenReview forum: "SemCEB: A Cardinality Estimation Benchmark for Semantic Operators"
_VLDB.org/2026/Workshop/NOVAS — NOVAS 2026_

### Official Review · Reviewer_4Eh9 · 2026-07-08

**Confidence:** 4

**Improvement Opportunities:**

- The main shortcoming in my opinion is that the evaluation is limited to a single dataset (Amazon products), in a limited subset (arts and crafts). While this is good to test the feasibility of methods, it doesn't help to determine whether the performance of a semantic operator generalizes for other domains. Expanding the dataset to a couple categories or sourcing additional catalogs will strengthen a lot this evaluation
- There is a section about embeddings, which are used to compute the topic distribution and the semantic skew, it is disconnected from the actual evaluation (section 4). I think that the subsections pertaining to the embeddings could either be integrated in a more useful way to the evaluation, for example, stratifying the performance of the semantic operators based off a subset determined by the inferred topics, or otherwise removed.
- Just one LLM (GPT-5.4) was used to conduct the evaluation. Ideally a few more should be used to compare the results. I understand it is expensive to run them, so take this suggestion in mind if this project grows in the future.

**Minor Comments:**

- It would be helpful to include a brief summary table describing the benchmark queries (e.g., modality, predicate category, selectivity range, and single vs. multi column) in the main paper rather than relying primarily on descriptive text. This would make the benchmark easier to understand and reproduce.

- The results should be included in a table. Box plots are nice to compare but vert hard to actually parse.

**Short Summary:**

This paper introduces SemCEB, a benchmark specifically designed to evaluate cardinality estimation methods for semantic database operators powered by LLMs. The authors argue that accurate cardinality estimation is a critical yet underexplored component of semantic query optimization because poor estimates can lead to significantly higher execution costs and latency when LLM-based operators are involved.
The paper presents an open-source benchmarking framework built on a multimodal Amazon Reviews dataset containing text, images, and precomputed embeddings, together with 102 manually curated semantic filter and join queries covering diverse predicate categories. The authors compare sampling-based estimators with an adaptation of Semantic Histograms. Their results show that sampling provides relatively robust accuracy across different predicate categories with substantial latency and inference cost, whereas Semantic Histograms are faster but have more limited applicability and exhibit reduced accuracy for certain predicate types.

**Strong Points:**

-  The introduction of SemCEB fills a gap in the literature by providing a benchmark specifically targeting semantic cardinality estimation.
-  The query workload spans multiple predicate categories (equality, negation, ordinal, temporal, spatial, and lexical), includes both single and multi column predicates.
-  In addition to q-error, the evaluation also considers practical deployment metrics including inference cost, latency, and memory usage. These additional metrics are particularly relevant for semantic query processing, where estimation itself may require expensive LLM calls and is limited by a budget in real life scenarios.

---

### Official Review · Reviewer_ofuz · 2026-07-11

**Confidence:** 4

**Improvement Opportunities:**

I really think this work is very important and what would have been nitpicks in my review (understandable given the space limitations) are discussed in the final section as limitations with plans of future work.

**Minor Comments:**

NA

**Short Summary:**

The authors introduce a first benchmark for cardinality estimation over semantic operators recently introduced in DBMSs.

**Strong Points:**

S1. Important problem since most semantic operator integrations assume zero cardinality estimate information during query planning and rely on run-time optimizations.

S2. Beyond the main contribution, the work also evaluates two cardinality estimation approaches.

S3. Clearly lays out limitations of the benchmark (hopefully the first of many in this space).

---

### Official Review · Reviewer_UiFM · 2026-07-14

**Confidence:** 3

**Improvement Opportunities:**

1. The stability of model-dependent ground truth should be treated as a first-class part of the benchmark protocol. The paper correctly observes that cardinalities may change with the model and prompt, but it does not sufficiently quantify stability for a fixed model–prompt configuration under stochastic decoding, repeated execution, or provider-side model updates.

**Minor Comments:**

The BERTopic/UMAP/HDBSCAN analysis of semantic skew is informative, but it depends on the selected embedding and clustering hyperparameters; the paper should clarify that this analysis characterizes the workload rather than defining an immutable notion of “true” semantic skew.

**Short Summary:**

This paper introduces SemCEB, a cardinality-estimation benchmark for semantic filters and semantic joins. The benchmark is built on the Arts, Crafts and Sewing subset of Amazon Reviews 2023 and contains approximately 45K products and 936K reviews. Its workload comprises 102 manually written queries. Unlike conventional cardinality-estimation benchmarks, SemCEB treats ground truth as model- and prompt-dependent and evaluates not only q-error but also estimation cost, latency, and additional storage. The paper provides an initial evaluation using uniform sampling at several rates and an adapted implementation of Semantic Histograms.

**Strong Points:**

1. The paper identifies an important problem in semantic query optimization that has not yet been evaluated systematically. The motivating example in Section 1 shows virtual inference costs ranging from $0.119 to $2.869 across 96 semantically equivalent plans, a 24.1× difference.

2. The workload is considerably richer than a collection of simple natural-language filters.

3. The benchmark handles model dependence pragmatically and transparently.

---

### Decision · Program_Chairs · 2026-07-16

**Decision:**

Accept

**Comment:**

SemCEB fills an important gap by introducing a benchmark for cardinality estimation over semantic filters and joins. The benchmark includes a diverse manually curated workload and evaluates estimation accuracy together with cost, latency, and storage, providing a strong foundation for future research in semantic query optimization. We hope this work sparks further development of cardinality estimators and productive discussion at the workshop.